# Optimization and Comparative Study of Different Extraction Methods of Sixteen Fatty Acids of *Potentilla anserina* L. from Twelve Different Producing Areas of the Qinghai-Tibetan Plateau

**DOI:** 10.3390/molecules27175443

**Published:** 2022-08-25

**Authors:** Guangxiang Luan, Mei Yang, Xingmei Nan, Huiling Lv, Qi Liu, Yuwei Wang, Yongfang Li

**Affiliations:** 1Department of Pharmacy, Medical College, Qinghai University, Xining 810016, China; 2State Key Laboratory of Plateau Ecology and Agriculture, Qinghai University, Xining 810016, China

**Keywords:** *Potentilla anserina* L., fatty acids, supercritical fluid extraction (SFE), ultrasonic-assisted extraction (UAE), microwave-assisted extraction (MAE), response surface methodology (RSM)

## Abstract

In this study, supercritical fluid extraction (SFE), ultrasonic-assisted extraction (UAE), and microwave-assisted extraction (MAE) were applied to explore the most suitable extraction method for fatty acids of *Potentilla anseris* L. from 12 different producing areas of the Qinghai-Tibetan Plateau. Meanwhile, the important experimental parameters that influence the extraction process were investigated and optimized via a Box-Behnken design (BBD) for response surface methodology (RSM). Under optimal extraction conditions, 16 fatty acids of *Potentilla anserina* L. were analyzed via high-performance liquid chromatography (HPLC) with fluorescence detection, using 2-(4-amino)-phenyl-1-hydrogen-phenanthrene [9,10-d] imidazole as the fluorescence reagent. The results showed that the amounts of total fatty acids in sample 6 by applying SFE, UAE, and MAE were, respectively, 16.58 ± 0.14 mg/g, 18.11 ± 0.13 mg/g, and 15.09 ± 0.11 mg/g. As an environmental protection technology, SFE removed higher amounts of fatty acids than did MAE, but lower amounts of fatty acids than did UAE. In addition, the contents of the 16 fatty acids of *Potentilla anserina* L. from the 12 different producing areas Qinghai-Tibetan Plateau were significantly different. The differences were closely related to local altitudes and to climatic factors that corresponded to different altitudes (e.g., annual mean temperature, annual mean precipitation, annual evaporation, annual sunshine duration, annual solar radiation.). The temperature indices, photosynthetic radiation, ultraviolet radiation, soil factors, and other factors were different due to the different altitudes in the growing areas of *Potentilla anserina* L., which resulted in different nutrient contents.

## 1. Introduction

*Potentilla anserina* L. is a perennial herb plant that is dominant in alpine meadows [1,2]. As a folk food and a traditional Tibetan medicine, it was known as “ginseng fruit.” It is mainly distributed in Qinghai Province, the Tibet Autonomous Region, Gansu Province (Gannan Prefecture), and Sichuan Province (Aba Prefecture and Garze Prefecture) in China [3,4]. According to existing literature, the roots of *Potentilla anserina* L. have high levels of polysaccharides [5], flavonoids, phenolic compounds [6,7], triterpenes, triterpene glycosides [8,9], ellagic acid glycosides [10], and amino acids [3]. In addition, the roots of this plant have a favorable nutritional value, with the characteristics of high protein, high dietary fiber, richness in fatty acids, various mineral elements, and low sodium content [11]. The abundant and diverse active substances of *Potentilla anserina* L. contribute to a wide variety of biological activities, including anticariogenic effects, hepatoprotective abilities [1,10], immunomodulatory effects [5], anti-inflammatory impact [12], and the ability to ameliorate acute hypobaric hypoxia-induced brain impairment [1].

Fatty acids, which are compounds composed of carbon, hydrogen, and oxygen, are the main components of neutral fat, phospholipid, and glycolipid. As important functional compounds, fatty acids play critical roles in improving insulin resistance [13,14], preventing and treating cancer [15], and treating cardiovascular diseases [16]. At present, the main methods for extracting fatty acids are supercritical fluid extraction (SFE), ultrasonic-assisted extraction (UAE), and microwave-assisted extraction (MAE) [17]. SFE, which is safe and reliable, does not require organic solvents to protect fatty acids from contamination or to improve their biological activity and purity. The UAE and MAE methods are simple and operable. 

In 2008, the main contents of the free fatty acids in *Potentilla anserina* L. from Yushu Tibetan Autonomous Prefecture, Qinghai, obtained by distillation extraction with n-hexane, were α-linolenic acid (3490 µg/g), linolic acid (7351 µg/g), and palmitic acid (2262 µg/g) [18]. However, the content and composition of fatty acids extracted from *Potentilla anseris* L. by the SFE, UAE, and MAE methods have not been reported. 

The purpose of this study is to identify the most suitable method for extracting fatty acids of *Potentilla anseris* L. from 12 different producing areas of the Qinghai–Tibetan Plateau. The pivotal extraction parameters that influenced the experimental process were investigated and optimized via a Box-Behnken design (BBD) for response surface methodology (RSM). On this basis, 16 fatty acids of *Potentilla anserina* L. were analyzed via high-performance liquid chromatography (HPLC) with fluorescence detection, using 2-(4-amino)-phenyl-1-hydrogen-phenanthrene [9,10-d] imidazole as the fluorescence reagent.

## 2. Materials and Methods

### 2.1. Materials

Samples of *Potentilla anserina* L. from 12 different producing areas of the Qinghai–Tibetan Plateau were identified on the basis of plant appearance. The tuberous root of *Potentilla anserine* L. was used for the extraction of fatty acids in this study. All samples were propagated by cuttings in 2018; they were 3 years old at the time of collection. All relevant information about the *Potentilla anserina* L. samples, including sample codes, producing areas, coordinates, altitudes, and picking times, are displayed in Table 1.

### 2.2. Reagent and Instruments

Sixteen fatty acids standards—octanoic acid, capric acid, undecanoic acid, lauric acid, myristic acid, α-linolenic acid, linolic acid, pentadecanoic acid, palmitic acid, oleic acid, heptadecanoic acid, stearic acid, *n*-nonadecylic acid, arachidic acid, *n*-heneicosanoic acid, and behenic acid—were purchased from Sigma Reagent Co. (St. Louis, MO, USA). 2-(4-amino)-phenyl-1-hydrogen-phenanthrene [9,10-d] imidazole was synthesized in our laboratory, as described in our previous research paper [19]. All other analytical grade reagents were provided by Alltech Scientific (Beijing, China), unless otherwise stated.

A high-performance liquid chromatography (HPLC) system and Zorbax Stablebond XDB-C18, Zorbax StableBond SB-C18, Zorbax Stablebond Extend-C18, Zorbax C18, and Eclipse XDB-C18 were obtained from Agilent Technologies Co., Ltd., Palo Alto, CA, USA. Hypersil™ GOLD, Hypersil™ BDS C8, and Hypersil C18 were provided by Thermo Fisher Scientific, Waltham, MA, USA. We used the Agilent1260 series HPLC system equipped with an online degasser, a quaternary pump, an autosampler, a thermostat column compartment, and a fluorescence detection detector. 

### 2.3. Experimental Design and Data Analysis

The Box-Behnken design (BBD) for response surface methodology (RSM) was utilized as the observatory indicator to investigate the extraction rate of total fatty acids from *Potentilla anserina* L by the SFE, UAE, and MAE methods. Samples from Gannan Tibetan Autonomous Prefecture were selected as representative, and SFE, UAE and MAE were all developed in static mode. The key total fatty acids extraction parameters by SFE were optimized, as follows: X_a1_, extraction temperature (values = 30, 45, and 60 °C); X_a2_, extraction pressure (values = 30, 35, and 40 MPa); and X_a3_, extraction time (values = 1, 2, and 3 h). The important parameters affecting the total fatty acids extraction by UAE were optimized as follows: X_b1_, extraction temperature (values = 30, 40, and 50 °C); X_b2_, extraction time (values = 30, 60, and 90 min); and X_b3_, extraction volume (values = 50, 70, and 90 mL). The critical characteristics affecting the total fatty acids extraction by MAE were optimized as follows: X_c1_, extraction power (values = 300, 400, and 500 W); X_c2_, extraction time (values = 30, 60, and 90 s); and X_c3_, extraction times (values = 3, 5, and 7). The dependent variable (Y) was the total peak area of the fatty acids. The experimental designs for the SFE, UAE, and MAE methods are shown in Table 2, Table 3 and Table 4.

### 2.4. SFE, UAE, and MAE

The SFE of the total fatty acids from *Potentilla anserina* L was conducted with supercritical fluid extraction equipment (Applied Separations Inc, Allentown, PA, USA) [17,20,21]. A 10 g powder of *Potentilla anserina* L was accurately weighed in supercritical carbon dioxide fluid extraction equipment for 2.1 h. The extraction temperature was 47 °C, the extraction pressure was 36 MPa, and the flow of carbon dioxide was 40 L/h. Then, the extract was collected.

The UAE of the total fatty acids from *Potentilla anserina* L was carried out with the MEC-200SAH ultrasonic extraction instrument (Wuxi Instrument Manufacturing Co., LTD, Wuxi, China) [22,23]. A 10 g powder of *Potentilla anserina* L was accurately weighed in a 250 mL triangular flask; then, a 73 mL methanol-methylene chloride solution (1:3, *v*/*v*) was added. The ultrasonic treatment power was 600 W, ultrasonic extraction was conducted for 67 min at 42 °C with pH 6.5, and the liquid/solid ratio was 7.3:1 mL/g.

For the MAE of the total fatty acids from *Potentilla anserina* L. [24,25], 10 g powder of *Potentilla anserina* L. was accurately weighed and inserted with 50 mL n-hexane into a triangle bottle. The microwave treatment power was 420 W, with intermittent radiation five times; each radiation time was 63 s. After each radiation, the triangle bottle was cooled to room temperature with cold water and then transferred into a microwave oven for radiation and filtered with an extraction bottle. 

After SFE, UAE, and MAE, the extract of total fatty acids from *Potentilla anserina* L. filtrated and determined by HPLC. 

### 2.5. Quantitative Analysis of Individual Fatty Acids

The original solution was prepared by dissolving 30.95 mg of 2-(4-amino)-phenyl-1-hydrogen-phenanthrene [9,10-d] imidazole in 10 mL of 0.3 mol/L 4-dimethyl aminopyridine. The low concentration solution was obtained by diluting the original solution with acetonitrile. The 16 fatty acids mixed standard solutions (5 × 10^−3^ mol/L) were prepared in acetonitrile [19].

20 µL 5 × 10^−3^ mol/L fatty acids mixed standard solutions or extracts of total fatty acids from *Potentilla anserina* L, 150 µL 5 × 10^−3^ mol/L 2-(4-amino)-phenyl-1-hydrogen- phenanthrene [9,10-d] imidazole, and 30 μL 0.15 mol/L 1-ethyl-3-(3-dimethyl ammonium propyl)-carbonyl imine were combined in a 2 mL vial, then sealed. After reaction in a water bath at 80 °C for 45 min, a 0.600 mL acetonitrile solution was added; then, the diluted solution was syringe-filtered [19].

A satisfactory separation of individual fatty-acids derivatives was achieved on an Agilent 1260 Series HPLC system with Hypersil GOLD (4.6 mm × 250 mm, 5 μm). Solvent A was 5% acetonitrile in water, while solvent B consisted of acetonitrile. The gradient conditions were as follows: 0–15 min, 35%A–20%A; 15–30 min, 20%A–15%A; 30–50 min, 15%A–0%A; and 50–60 min, 0%A. The wavelengths of fluorescence detection were λ_ex_/λ_em_ = 262/424 nm.

## 3. Results and Discussion

### 3.1. Optimization of SFE, UAE, and MAE Conditions

In order to obtain the optimal extraction conditions, a series of extraction variables was designed to optimize and explore the interactions between these variables by the BBD. The experimental design and results of fatty acids were provided in Table 2, Table 3 and Table 4. 

In optimizing the SFE conditions, the F-value of the model was 91.86, demonstrating that the model was significant. The “prob > F” values < 0.0500 implied that the model terms were significant, including X_a1_, X_a2_, X_a3_, X_a1_X_a2_, X_a1_X_a3_, X_a2_X_a3_, X_a1_^2^, X_a2_^2^, and X_a3_^2^. The results indicated that extraction temperature, extraction pressure, and extraction time were the most important parameters affecting SFE efficiency, and the quadratic model could describe precisely the experimental response. The final equation (Equation (1)) for the SFE design is shown below:Y_1_ = 3904.00 + 72.38 X_a1_ + 64.63 X_a2_ + 64.75 X_a3_ − 110.50 X_a1_X_a2_ − 166.25 X_a1_X_a3_ − 127.75 X_a2_X_a3_ − 142.75 X_a1_^2^ − 157.75 X_a2_^2^ − 194.50 X_a3_^2^(1)

The three-dimensional response surfaces (Figure 1a–c) are represented on the basis of the optimal conditions, and the interaction between the variables was investigated to determine the optimization of the maximum content of the fatty acids. Figure 1a demonstrates the combined effects of extraction temperature and extraction pressure. Figure 1b highlights the combined effects of extraction temperature and extraction time. Figure 1c depicts the combined effects of extraction pressure and extraction time. Based on the overall results of the optimization study with the actual convenience of the experimental operation, the optimal SFE conditions were selected as follows: extraction temperature = 47 °C, extraction pressure = 36 MPa, and extraction time = 2.1 h. Under these optimal conditions, the peak area of the fatty acids was 3915.989.

In optimizing the UAE conditions, the F-value of the model was 88.40, demonstrating that the model was significant. The “prob > F” values < 0.0500 implied that the model terms were significant, including X_b1_, X_b2_, X_b3_, X_b1_X_b2_, X_b1_^2^, X_b2_^2^, and X_b3_^2^. The results indicated that extraction temperature, extraction time, and extraction volume were the most important parameters affecting UAE efficiency, and the quadratic model could describe precisely the experimental response. The final equation (Equation (2)) for the UAE design is shown below:Y_2_ = 4176.00 + 111.38 X_b1_ + 105.50 X_b2_ + 82.13 X_b3_ − 42.50 X_b1_X_b2_ + 1.75 X_b1_X_b3_ − 39.00 X_b2_X_b3 −_ 225.13 X_b1_^2^ − 203.87 X_b2_^2^ − 227.63 X_b3_^2^(2)

The three-dimensional response surfaces (Figure 1d–f) are represented on the basis of the optimal conditions, and the interaction between the variables was investigated to determine the optimization of the maximum content of the fatty acids. Figure 1d demonstrates the combined effects of extraction temperature and extraction time. Figure 1e highlighted the combined effects of extraction temperature and extraction volume. Figure 1f depicts the combined effects of extraction time and extraction volume. Based on the overall results of the optimization study with the actual convenience of the experimental operation, the optimal SFE conditions were selected as follows: extraction temperature = 42 °C, extraction time = 67 min, and extraction volume = 73 mL. Under these optimal conditions, the peak area of the fatty acids was 4206.907.

In optimizing the MAE conditions, the F-value of the model was 14.66, demonstrating that the model was significant. The “prob > F” values < 0.0500 implied that the model terms were significant, including Xc_1_, Xc_3_, Xc_1_^2^, Xc_2_^2^, and Xc_3_^2^. The results indicated that extraction power and extraction volume were the most important parameters affecting MAE efficiency, and the quadratic model could describe precisely the experiment response. The final equation (Equation (3)) for the MAE design is shown below:Y_3_ = 3607.00 + 82.25 X_c1_ + 52.13 X_c2_ + 65.63 X_c3_ − 28.50 X_c1_X_c2_ − 53.00 X_c1_X_c3_ − 23.25 X_c2_X_c3_ − 193.38 X_c1_^2^ − 220.6 2X_c2_^2^ − 225.13 X_c3_^2^(3)

The three-dimensional response surfaces (Figure 1g–i) are represented on the basis of the optimal conditions, and the interaction between the variables was investigated to determine the optimization of the maximum content of the fatty acids. Figure 1g demonstrates the combined effects of extraction power and extraction time. Figure 1h highlights the combined effects of extraction power and extraction times. Figure 1f depicts the combined effects of extraction time and extraction times. Based on the overall results of the optimization study with the actual convenience of the experimental operation, the optimal MAE conditions were selected as follows: extraction power = 420 W, extraction time = 63 s, and extraction times = 5. Under these optimal conditions, the peak area of the fatty acids was 3621.258.

### 3.2. Optimization of HPLC Separation

In order to obtain the best HPLC separation conditions of SFE, UAE, and MAE, chromatographic columns were compared, including Hypersil™ GOLD, Hypersil™ BDS C8, Hypersil C18, Zorbax Stablebond XDB-C18, Zorbax StableBond SB-C18, Zorbax Stablebond Extend-C18, Zorbax C18, and Eclipse XDB-C18. According to the HPLC chromatograms, Hypersil™ GOLD (250 mm × 4.6 mm, 5 μm) had the most suitable separation efficiency and a more symmetric separation peak was obtained. The representative chromatograms of blank, fatty acids standard solutions, and real samples are shown in Figure 2.

### 3.3. Validation of the Method

The optimized analysis method for the fatty acids was validated by a linear regression equation, limits of detection (LODs), limits of quantification (LOQs), and intra-day and inter-day precisions. The linearity relationships were provided by the plot of peak area versus the amounts of the sixteen fatty acids standards. As summarized in Table 5, the correlation coefficients of octanoic acid, capric acid, undecanoic acid, lauric acid, myristic acid, α-linolenic acid, linolic acid, pentadecanoic acid, palmitic acid, oleic acid, heptadecanoic acid, stearic acid, *n*-nonadecylic acid, arachidic acid, *n*-heneicosanoic acid and behenic acid were higher than 0.9960, with excellent linear responses. In addition, the LOD and LOQ ranges were from 0.14 ng/mL to 1.37 ng/mL and 1.18 ng/mL to 3.40 ng/mL, respectively, and the instrument precision of the intra-day and inter-day validations was <2.07 and 2.19, respectively. As summarized in Table 6, the percentage recoveries ranged from 97.0% to 103.0%, calculated by the ratio of the spiked samples concentrations to the actual samples concentrations. These results clearly indicated that the optimized method was precise and suitable for analysis of the 16 fatty acids in *Potentilla anserina* L. from 12 different producing areas of the Qinghai-Tibetan Plateau. 

### 3.4. Comparison of SFE, UAE and MAE

Considering *Potentilla anserina* L. from Gannan Tibetan Autonomous Prefecture as an example, the amounts of total fatty acids by SFE, UAE, and MAE were 16.58 ± 0.14 mg/g, 18.11 ± 0.13 mg/g, and 15.09 ± 0.11 mg/g, respectively. As Table 7, Table 8 and Table 9 show, the amounts of total fatty acids in samples from Yushu Tibetan Autonomous Prefecture Qinghai by SFE, UAE, and MAE were 14.08 ± 0.11 mg/g, 15.13 ± 0.11 mg/g, and 12.67 ± 0.10 mg/g, respectively. As a safety and environmental protection technology for extracting the 16 fatty acids of *Potentilla anserina* L. from 12 different producing areas of the Qinghai–Tibetan Plateau, SFE removed higher amounts of fatty acids than did MAE, but lower amounts of fatty acids than did UAE. Overall, there was no need to introduce organic solvents in the experimental process to protect fatty acids from pollution or to preserve their high biological activity and purity. UAE was operable and yielded the highest fatty acids with a simple test device. Compared with the amounts of fatty acids obtained by SFE and UAE, the amount of fatty acids obtained by MAE was lower, but required shorter time (the extraction times for MAE, UAE, and SFE were 63 s, 67 min, and 2.1 h, respectively).

In addition, in contrast with other studies that used different methods of free fatty acids extraction [18], the total content of the free fatty acids in *Potentilla anserina* L. from Yushu Tibetan Autonomous Prefecture Qinghai, by distillation extraction with *n*-hexane, was 14.427 mg/g. In this study, the total contents of the free fatty acids in same sample by SFE, UAE and MAE were 14.08 ± 0.11 mg/g, 15.13 ± 0.11 mg/g, and 12.67 ± 0.10 mg/g, respectively. These results show that there are certain differences in the contents of the fatty acids obtained from the same sample by distillation extraction with n-hexane, UAE, and MAE. It demonstrates that it is crucial to explore the most suitable method for extracting the fatty acids of *Potentilla anseris* L.

### 3.5. Analysis of 16 Fatty Acids of Potentilla anserina L. from 12 Different Producing Areas of the Qinghai–Tibetan Plateau 

The established SFE, UAE, and MAE methods and the optimal conditions were applied to analyze the fatty acids of *Potentilla anserina* L. The composition data of the fatty acids by the three extraction methods from the dry *Potentilla anserina* L. are shown in Table 7, Table 8 and Table 9.

Based on the experimental study results, the contents of the 16 fatty acids of *Potentilla anserina* L. from 12 different producing areas of the Qinghai-Tibetan Plateau were significantly different. Considering UAE, for example, the amount of total fatty acids from sample 6 was high, up to 18.11 ± 0.13 mg/g, while the amount of total fatty acids from sample 11 was only 10.11 ± 0.08 mg/g. Moreover, the contents of linolic acid, oleic acid, octanoic acid, and palmitic acid were highest from *Potentilla anserina* L. The contents of linolic acid were 6092.60 ± 37.70 µg/g in sample 6, 5506.50 ± 36.90 ug/g in sample 1, 4549.26 ± 36.40 µg/g in sample 4, and 4350.08 ± 35.22 µg/g in sample 5. In addition, linolic acid has a prevention-and-treatment function for many conditions or diseases, including obesity, cancer, diabetes, and cardiovascular diseases [26]. The fatty acids of *Potentilla anserina* L. were quite different in different producing areas. As shown in Table 7, Table 8 and Table 9, the total fatty-acids contents of *Potentilla anserina* L. growing in Gannan Tibetan Autonomous Prefecture was high, 18.11 ±0.13 mg/g, and the total fatty-acids contents of samples growing in Lhatse County Tibet Autonomous Region, based on UAE, was only 10.11 ± 0.08 mg/g. As shown in Table 1, the altitudes of Gannan Tibetan Autonomous Prefecture and Anduo County Tibet Autonomous Region are 2950 m and 4600 m, respectively, and the coordinates of Gannan Tibetan Autonomous Prefecture and Anduo County Tibet Autonomous Region are 33°06′37.49′′ N, 100°46′12.77′′ E and 31°40′14.31′′ N, 91°51′23.78′′ E, respectively. The samples were planted artificially, so the factors that can be controlled by researchers were basically the same. The differences in the fatty acids of *Potentilla anserina* L. from different habitats maybe related to local altitudes. 

## 4. Conclusions

In this study, supercritical fluid extraction (SFE), ultrasonic-assisted extraction (UAE), and microwave-assisted extraction (MAE) were applied to explore the most suitable extraction method for 16 fatty acids of *Potentilla anseris* L. from 12 different producing areas of the Qinghai–Tibetan Plateau. Under optimal extraction conditions, the 16 fatty acids of *Potentilla anserina* L.—octanoic acid, capric acid, undecanoic acid, lauric acid, myristic acid, α-linolenic acid, linolic acid, pentadecanoic acid, palmitic acid, oleic acid, heptadecanoic acid, stearic acid, *n*-nonadecylic acid, arachidic acid, *n*-heneicosanoic acid, and behenic acid—were analyzed by HPLC with fluorescence detection, using 2-(4-amino)-phenyl-1-hydrogen-phenanthrene [9,10-d] imidazole as the fluorescence reagent. The results showed that the amounts of total fatty acids in sample 6 by applying SFE, UAE, and MAE, respectively, were 16.58 ± 0.14 mg/g, 18.11 ± 0.13 mg/g, and 15.09 ± 0.11 mg/g, and the amounts of total fatty acids in sample 1 by applying SFE, UAE, and MAE, respectively, were 14.08 ± 0.11 mg/g, 15.13 ± 0.11 mg/g, and 12.67 ± 0.10 mg/g. As an environmental protection technology, SFE removed higher amounts of fatty acids than did MAE, but lower amounts of fatty acids than did UAE. UAE was operable and had the highest fatty acids yield, with a simple testing device. Compared with the amounts of fatty acids obtained by SFE and UAE, the amount of fatty acids obtained by MAE was lower, but the method required shorter time. In addition, the contents of the 16 fatty acids of *Potentilla anserina* L. from the 12 different producing areas of the Qinghai–Tibetan Plateau were significantly different. Based on UAE, for example, the amount of total fatty acids from sample 6 was high, 18110.23 ± 128.92 µg/g, while the amount of total fatty acids from sample 11 was only 10110.06 ± 81.16 µg/g. The differences were closely related to local altitudes and to climatic factors that corresponded to different altitudes (e.g., annual mean temperature, annual mean precipitation, annual evaporation, annual sunshine duration, and annual solar radiation). Because of the different altitudes, the temperature indices, photosynthetic radiation, ultraviolet radiation, soil factors, and other factors were different in the growing area of *Potentilla anserina* L., resulting in different nutrient contents. 

## Figures and Tables

**Figure 1 molecules-27-05443-f001:**
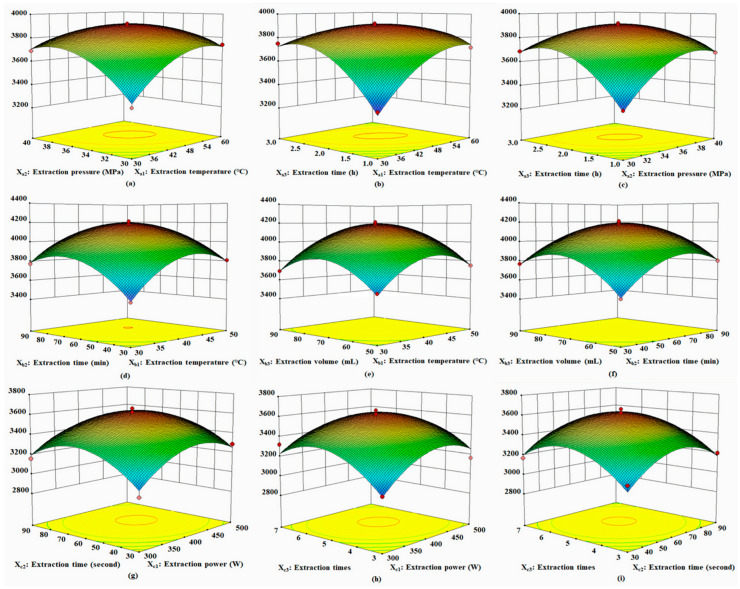
3D surface plot showing the significant interaction effects of the extraction parameters: (**a**–**c**) demonstrates the combined effects of SFE; (**d**–**f**) highlights the combined effects of UAE; (**g**–**i**) depicts the combined effects of MAE.

**Figure 2 molecules-27-05443-f002:**
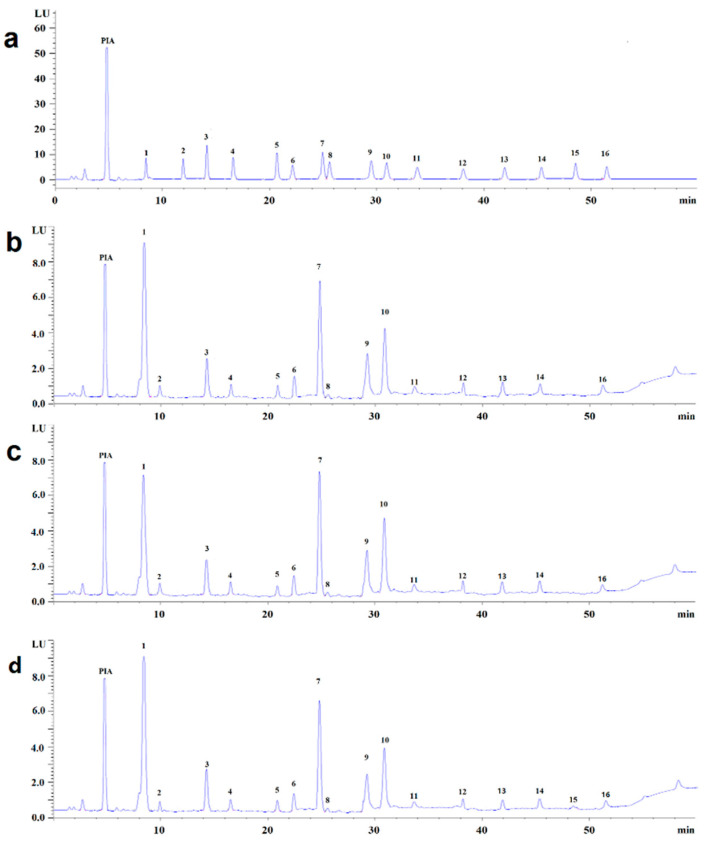
The representative chromatograms for standards (**a**), the typical chromatograms for fatty acid extracts in *Potentilla anserina* L from Gannan Tibetan Autonomous Prefecture by SFE (**b**), the typical chromatograms for fatty acid extracts in *Potentilla anserina* L from Gannan Tibetan Autonomous Prefecture by UAE (**c**), and the typical chromatograms for fatty acid extracts in *Potentilla anserina* L. from Gannan Tibetan Autonomous Prefecture by MAE (**d**). Peak labels are 1 for octanoic acid, 2 for capric acid, 3 for undecanoic acid, 4 for lauric acid, 5 for myristic acid, 6 for α-linolenic acid, 7 for linolic acid, 8 for pentadecanoic acid, 9 for palmitic acid, 10 for oleic acid, 11 for heptadecanoic acid, 12 for stearic acid, 13 for *n*-nonadecylic acid, 14 for arachidic acid, 15 for *n*-heneicosanoic acid, and 16 for behenic acid. PIA: 2-(4-amino)-phenyl-1-hydrogen-phenanthrene [9,10-d] imidazole.

**Table 1 molecules-27-05443-t001:** Information about the *Potentilla anserina* L. samples used in this study.

Sample Codes	Producing Areas	Coordinates	Altitude	Picking Time
Sample 1	Yushu Tibetan Autonomous Prefecture Qinghai	32°04′48.77′′ N 96°50′51.63′′ E	3700	10 June 2021
Sample 2	Guoluo Tibetan Autonomous Prefecture Qinghai	33°15′19.17′′ N 100°16′35.72′′ E	4300	21 June 2021
Sample 3	Hainan Tibetan Autonomous Prefecture Qinghai	35°53′9.34′′ N 100°51′50.11′′ E	2800	12 June 2021
Sample 4	Haibei Tibetan Autonomous Prefecture Qinghai	37°22′34.78′′ N 101°37′25.91′′ E	2950	24 June 2021
Sample 5	Haixi Mongolian and Tibetan Autonomous Prefecture Qinghai	36°25′27.28′′ N 98°09′2.36′′ E	2800	16 June 2021
Sample 6	Gannan Tibetan Autonomous Prefecture	33°06′37.49′′ N 100°46′12.77′′ E	2950	6 June 2021
Sample 7	Nagqu County Tibet Autonomous Region	31°28′15.57′′ N 90°6′12.38′′ E	4600	4 June 2021
Sample 8	Anduo County Tibet Autonomous Region	31°40′14.31′′ N 91°51′23.78′′ E	4600	11 June 2021
Sample 9	Maqen County Tibet Autonomous Region	34°51′37.52′′ N 100°07′44.91′′ E	4100	19 June 2021
Sample 10	Nyingchi County Tibet Autonomous Region	29°33′72.68′′ N 94°21′66.33′′ E	3050	26 June 2021
Sample 11	Lhatse County Tibet Autonomous Region	29°15′26.18′′ N 88°07′33.21′′ E	4150	19 June 2021
Sample 12	MarkamTibet Autonomous Region	29°60′27.27′′ N 98°58′69.20′′ E	3800	26 June 2021

**Table 2 molecules-27-05443-t002:** The experimental design and data for SFE, obtained via BBD for fatty acids (*n* = 3).

Run	Extraction Temperature (°C)	Extraction Pressure (MPa)	Extraction Time (h)	Peak Area
1	30	35	3	3751
2	45	30	1	3304
3	45	35	2	3912
4	60	30	2	3738
5	45	30	3	3687
6	45	40	1	3672
7	45	35	2	3887
8	30	35	1	3287
9	45	35	2	3896
10	30	30	2	3323
11	60	35	1	3715
12	45	40	3	3544
13	60	40	2	3663
14	60	35	3	3514
15	30	40	2	3690
16	45	35	2	3921
17	45	35	2	3904

**Table 3 molecules-27-05443-t003:** The experimental design and data for UAE, obtained via BBD for fatty acids (*n* = 3).

Run	Extraction Temperature (°C)	Extraction Time (min)	Extraction Volume (mL)	Peak Area
1	40	60	70	4179
2	50	60	50	3750
3	50	30	70	3807
4	50	90	70	3930
5	40	90	50	3799
6	30	60	50	3551
7	50	60	90	3899
8	40	90	90	3904
9	30	90	70	3772
10	30	60	90	3693
11	40	30	90	3768
12	40	60	70	4187
13	40	60	70	4106
14	40	60	70	4198
15	40	60	70	4210
16	40	30	50	3507
17	30	30	70	3479

**Table 4 molecules-27-05443-t004:** The experimental design and data for MAE, obtained via BBD for fatty acids (*n* = 3).

Run	Extraction Power (W)	Extraction Time (s)	Extraction Times	The Peak Area
1	500	90	5	3362
2	400	60	5	3657
3	300	30	5	2967
4	500	60	7	3279
5	400	60	5	3586
6	400	60	5	3618
7	500	60	3	3165
8	300	90	5	3157
9	500	30	5	3286
10	300	60	7	3318
11	400	60	5	3570
12	400	30	7	3168
13	400	60	5	3604
14	400	90	3	3201
15	400	90	7	3197
16	300	60	3	2992
17	400	30	3	3079

**Table 5 molecules-27-05443-t005:** Linear regression equation, correlation coefficients, limits of detection (LODs), limits of quantification (LOQs), reproducibility of retention time and the peak area, and intra- and inter-day precisions.

Analyte	Regression Equation	r	LOD (µg/L)	LOQ (µg/L)	Instrument Precision (*n* = 6)		Method Precision (*n* = 3)	
					Intra-Day	Inter-Day	Intra-Day	Inter-Day
octanoic acid	y = 0.341x + 1.937	0.9976	0.14	1.18	0.68	0.86	1.22	2.67
capric acid	y = 0.406x + 0.528	0.9990	0.15	2.60	0.57	0.92	1.21	2.30
undecanoic acid	y = 0.609x − 4.213	0.9973	0.21	1.40	0.71	1.06	1.53	2.67
lauric acid	y = 0.427x − 0.301	0.9986	0.34	1.69	0.69	0.89	1.25	2.42
myristic acid	y = 0.608x − 0.547	0.9987	0.32	2.90	1.03	1.40	2.06	3.91
pentadecanoic acid	y = 0.371x − 1.507	0.9971	0.34	1.70	0.57	0.87	1.19	2.72
palmitic acid	y = 0.814x − 1.592	0.9977	1.37	3.40	2.07	2.19	3.70	1.54
heptadecanoic acid	y = 0.567x + 0.492	0.9985	0.58	1.87	0.70	0.87	2.24	3.70
stearic acid	y = 0.674x − 1.556	0.9986	1.09	2.38	1.56	2.00	3.53	2.34
*n*-nonadecylic acid	y = 0.542x + 7.714	0.9992	1.60	2.97	1.62	2.06	4.27	1.81
arachidic acid	y = 0.496x + 0.680	0.9979	0.31	1.79	0.86	1.31	1.73	3.69
*n*-heneicosanoic acid	y = 0.425x − 1.482	0.9980	0.37	2.00	0.90	1.44	1.69	2.90
behenic acid	y = 0.428x − 1.532	0.9990	0.44	2.03	0.93	1.52	1.58	3.25
oleic acid	y = 0.522x − 4.348	0.9978	0.31	1.68	0.76	1.36	1.50	2.71
linolic acid	y = 0.588x − 1.255	0.9988	0.39	1.51	1.28	1.72	2.66	5.62
α-Linolenic acid	y = 0.556x − 2.000	0.9960	0.40	1.78	1.07	1.36	1.61	4.13

**Table 6 molecules-27-05443-t006:** Recovery studies of the proposed method at three concentration levels.

Analyte	Concentration 1			Concentration 2			Concentration 3		
	Added	Found	Recovery	Added	Found	Recovery	Added	Found	Recovery
	(µg/L)	(µg/L)	(%)	(µg/L)	(µg/L)	(%)	(µg/L)	(µg/L)	(%)
octanoic acid	0.50	0.50	100	1.00	1.01	101	2.00	2.02	101.1
capric acid	0.50	0.51	102	1.00	0.99	99	2.00	1.98	99.3
undecanoic acid	0.50	0.49	98	1.00	1.01	101	2.00	2.00	100
lauric acid	0.50	0.49	98	1.00	0.99	99	2.00	1.99	99.5
myristic acid	0.50	0.50	100	1.00	0.97	97	2.00	2.00	100
pentadecanoic acid	0.50	0.51	102	1.00	1.01	101	2.00	1.98	99.3
palmitic acid	0.50	0.50	100	1.00	0.99	99	2.00	1.97	98.5
heptadecanoic acid	0.50	0.49	98	1.00	0.98	98	2.00	2.00	100
stearic acid	0.50	0.51	102	1.00	1.02	102	2.00	2.00	100
*n*-nonadecylic acid	0.50	0.50	100	1.00	1.00	100	2.00	2.00	100
arachidic acid	0.50	0.49	98	1.00	1.01	101	2.00	1.97	98.5
*n*-heneicosanoic acid	0.50	0.51	102	1.00	0.99	99	2.00	2.00	100
behenic acid	0.50	0.51	102	1.00	1.03	103	2.00	1.99	99.5
oleic acid	0.50	0.50	100	1.00	0.97	97	2.00	2.00	100
linolic acid	0.50	0.50	100	1.00	0.99	99	2.00	1.99	99.5
α-Linolenic acid	0.50	0.49	98	1.00	1.00	100	2.00	2.02	101.1

**Table 7 molecules-27-05443-t007:** The contents of fatty acids from *Potentilla anserina* L. by SFE (µg/g, *n* = 3).

Samples	Sample 1	Sample 2	Sample 3	Sample 4	Sample 5	Sample 6
octanoic acid	2163.81 ± 16.52	2213.73 ± 16.95	2031.17 ± 15.01	2771.31 ± 17.87	2936.02 ± 19.68	3008.56 ± 21.09
capric acid	39.33 ± 0.28	47.56 ± 0.67	-	32.40 ± 0.23	43.41 ± 0.27	71.37 ± 0.67
undecanoic acid	1186.79 ± 9.55	1425.98 ± 12.80	1227.37 ±10.23	1218.33 ± 10.67	1382.25 ± 11.20	1327.87 ± 12.07
lauric acid	20.87 ± 0.23	36.60 ± 0.31	45.69 ± 0.42	41.35 ± 0.48	51.17 ± 0.63	76.06 ± 0.57
myristic acid	30.12 ± 0.83	-	27.65 ± 0.15	43.47 ± 0.32	52.41 ± 0.42	63.65 ± 0.56
α-Linolenic acid	637.04 ± 0.52	611.43 ± 5.32	582.84 ± 4.70	507.94 ± 4.51	316.79 ± 3.05	778.55± 6.82
linolic acid	5135.62 ± 38.26	5017.79 ± 36.94	4873.62± 37.67	4211.81 ± 35.74	5215.25 ± 39.69	5619.36 ± 50.04
pentadecanoic acid	-	32.27 ± 0.42	49.36 ± 0.64	81.76 ± 0.81	40.51 ± 0.48	64.48 ± 0.52
palmitic acid	1676.33 ± 13.14	1477.62 ± 12.22	1313.36 ± 11.23	1337.23 ± 11.08	1267.73 ± 10.06	1400.53 ± 12.64
oleic acid	3017.82 ± 26.76	2636.98 ± 20.86	2624.48 ± 21.47	1728.89 ± 13.09	2201.47 ± 20.45	3702.97 ± 30.62
heptadecanoic acid	44.43 ± 2.81	46.36 ± 0.43	41.16± 0.40	43.69 ± 0.13	38.12 ± 0.33	72.16 ± 0.60
stearic acid	49.31 ± 0.28	71.50 ± 0.51	67.84 ± 0.71	81.84 ± 0.72	38.39 ± 0.34	168.49± 0.82
*n*-nonadecylic acid	-	-	35.63± 0.21	47.53 ± 0.29	-	80.01 ± 0.67
arachidic acid	-	62.26 ± 0.52	-	-	52.22 ± 0.47	75.68 ± 0.80
*n*-heneicosanoic acid	51.91 ± 0.43	61.18 ± 0.70	37.42 ± 0.25	47.07 ± 0.37	72.37 ± 0.61	-
behenic acid	22.73 ± 0.23	27.00 ±0.23	38.33 ± 0.33	-	-	67.29 ± 0.54
The total readings (mg/g)	14.08 ± 0.11	13.77 ± 0.11	13.00 ± 0.10	12.19 ± 0.10	13.71 ± 0.11	16.58 ± 0.14
Samples	Sample 7	Sample 8	Sample 9	Sample 10	Sample 11	Sample 12
octanoic acid	2013.17 ± 15.65	1830.17 ± 12.31	2351.25 ± 17.41	2137.25 ± 16.28	2163.05 ± 16.12	1951.08 ± 12.47
capric acid	-	51.51 ± 0.47	37.25 ± 0.28	31.62 ± 0.32	-	49.99 ± 0.44
undecanoic acid	1239.37 ±10.24	1411.18 ± 13.87	1062.15 ± 9.43	986.20 ± 8.23	1271.13 ± 12.31	1072.63 ± 11.61
lauric acid	46.57 ± 0.42	22.44 ± 0.25	46.24 ± 0.35	50.71 ± 0.40	64.32 ± 0.57	41.54 ± 0.35
myristic acid	20.65 ± 0.15	26.36 ± 0.22	73.64 ± 0.64	33.05 ± 0.32	56.11 ± 0.46	85.12 ± 0.77
α-Linolenic acid	567.68 ± 4.70	612.45 ± 5.95	536.51 ± 5.16	424.24 ± 3.87	645.23 ± 5.84	526.41 ± 4.51
linolic acid	4073.62± 37.67	1213.56 ± 10.97	2602.20 ± 18.72	2710.30 ± 18.35	3615.76 ± 24.42	4414.51 ± 33.99
pentadecanoic acid	52.45 ± 0.64	34.48 ± 0.32	40.21 ± 0.37	39.68 ± 0.32	65.26 ± 0.54	22.10 ± 0.16
palmitic acid	1283.36 ± 11.23	1277.86 ± 11.56	972.07 ± 8.05	1076.43 ± 9.41	891.78 ± 7.75	1115.76 ± 10.74
oleic acid	2135.48 ± 18.78	2106.69 ± 18.58	2201.20 ± 21.50	1396.51 ± 12.50	1752.41 ± 15.09	2212.21 ± 20.28
heptadecanoic acid	41.16± 0.40	51.08 ± 0.46	67.46 ± 0.65	54.36 ± 0.44	-	42.09 ± 0.37
stearic acid	67.84 ± 0.71	-	33.00 ± 0.24	37.14 ± 0.33	25.88 ± 0.26	53.44 ± 0.42
*n*-nonadecylic acid	27.63± 0.21	28.20 ± 0.23	51.06 ± 0.44	50.50 ± 0.45	27.63 ± 0.16	42.22 ± 0.37
arachidic acid	21.38 ± 0.37	-	35.32 ± 0.29	-	41.23 ± 0.39	33.63 ± 0.28
*n*-heneicosanoic acid	41.42 ± 0.25	35.27 ± 0.34	41.97 ± 0.27	-	33.83 ± 0.30	37.25 ± 0.30
behenic acid	45.33 ± 0.43	-	58.21 ± 0.49	-	40.02 ± 0.35	56.98 ± 0.41
The total readings(mg/g)	11.68 ± 0.10	8.70 ± 0.08	10.21 ± 0.08	9.03 ± 0.07	10.69 ± 0.08	11.76 ± 0.10

Data are expressed as mean value ± S.D. -. Not detected.

**Table 8 molecules-27-05443-t008:** The contents of fatty acids from *Potentilla anserina* L. by UAE (µg/g, *n* = 3).

Samples	Sample 1	Sample 2	Sample 3	Sample 4	Sample 5	Sample 6
octanoic acid	2353.36 ± 16.10	3984.40 ± 31.54	2187.32 ± 15.66	2913.34 ± 18.97	5182.54 ± 39.50	3379.61 ± 23.25
capric acid	41.77 ± 0.35	85.60 ± 0.71	-	34.59 ± 0.36	77.66 ± 0.70	57.08 ± 0.48
undecanoic acid	1235.75 ± 11.23	1566.56 ± 13.14	1121.73 ± 8.90	1625.94 ± 14.63	1172.84 ± 8.75	1434.26 ± 12.53
lauric acid	22.69 ± 0.18	65.87 ± 0.55	49.20 ± 0.43	40.66 ± 0.33	91.94 ± 0.83	87.05 ± 0.64
myristic acid	36.75 ± 0.29	-	21.67 ± 0.16	46.65 ± 0.43	93.76 ± 0.81	68.75 ± 0.40
α-Linolenic acid	626.84 ± 5.75	1100.48 ± 9.68	605.64 ± 5.79	508.63 ± 4.66	516.73 ± 4.57	877.93 ± 6.50
linolic acid	5506.50 ± 36.90	3231.32 ± 22.60	4135.30 ± 34.64	4549.26 ± 36.40	4350.08 ± 35.22	6092.60 ± 37.70
pentadecanoic acid	37.11 ± 0.28	58.08 ± 0.51	53.15 ± 0.55	88.31 ± 0.71	72.47 ± 0.60	61.64 ± 0.46
palmitic acid	1807.17 ± 11.7	2652.70 ± 18.59	1204.33 ± 9.71	1467.36 ± 12.79	2037.96 ± 14.69	1592.74 ± 11.85
oleic acid	3282.18 ± 22.81	3746.19 ± 28.65	2826.25 ± 20.50	1453.40 ± 12.59	2938.43 ± 18.40	3989.66 ± 31.63
heptadecanoic acid	48.32 ± 0.32	81.74 ± 0.72	41.72 ± 0.44	47.19 ± 0.40	38.19 ± 0.37	77.94 ± 0.62
stearic acid	53.19 ± 0.50	102.69 ± 0.90	73.05 ± 0.66	88.33 ± 0.77	66.68 ± 0.50	181.09 ± 1.20
*n*-nonadecylic acid	-	-	31.35 ± 0.29	51.33 ± 0.48	-	85.42 ± 0.71
arachidic acid	-	143.05 ± 1.29	-	-	91.12 ± 0.72	71.74 ± 0.54
*n*-heneicosanoic acid	56.45 ± 0.57	160.11 ± 1.35	49.29 ± 0.47	57.84 ± 0.51	118.27 ± 0.90	-
behenic acid	24.09 ± 0.21	41.89 ± 0.36	41.67 ± 0.37	-	-	52.68 ± 0.41
The total readings(mg/g)	15.13 ± 0.11	17.02 ± 0.13	12.44 ± 0.10	12.99 ± 0.10	16.85 ± 0.13	18.11 ± 0.13
Samples	Sample 7	Sample 8	Sample 9	Sample 10	Sample 11	Sample 12
octanoic acid	2165.43 ± 16.92	1971.21 ± 14.68	2452.17 ± 18.92	2290.03 ± 18.50	2375.79 ± 19.01	2101.07 ± 16.07
capric acid	-	45.62 ± 0.40	40.27 ± 0.35	54.47 ± 0.42	-	54.55 ± 0.41
undecanoic acid	1338.26 ± 10.63	1556.79 ± 11.82	1178.39 ± 9.77	1062.15 ± 8.75	1072.64 ± 8.62	1170.48 ± 9.96
lauric acid	55.32 ± 0.44	44.23 ± 0.41	50.99 ± 0.45	51.68 ± 0.44	109.45 ± 0.97	45.33 ± 0.40
myristic acid	26.21 ± 0.24	38.46 ± 0.34	71.62 ± 0.50	32.43 ± 0.31	80.59 ± 0.61	92.88 ± 0.85
α-Linolenic acid	623.13 ± 5.65	641.32 ± 5.74	575.07 ± 4.85	436.50 ± 3.76	706.75 ± 5.88	583.43 ± 4.44
linolic acid	4301.95 ± 34.44	2322.40 ± 19.72	2662.09 ± 21.69	3757.36 ± 24.69	2664.93 ± 20.35	3470.32 ± 22.76
pentadecanoic acid	56.67 ± 0.47	37.23 ± 0.32	43.47 ± 0.35	43.25 ± 0.39	71.47 ± 0.62	94.11 ± 0.86
palmitic acid	1306.79 ± 10.19	1337.83 ± 10.33	1107.00 ± 9.62	1203.52 ± 10.04	932.99 ± 8.58	1197.55 ± 10.01
oleic acid	2207.60 ± 17.50	2221.80 ± 16.94	2313.93 ± 17.37	1537.47 ± 12.65	1869.35 ± 14.67	2384.03 ± 19.70
heptadecanoic acid	43.49 ± 0.37	45.15 ± 0.46	72.43 ± 0.68	51.26 ± 0.43	-	75.93 ± 0.60
stearic acid	69.32 ± 0.58	-	31.18 ± 0.25	43.49 ± 0.40	47.94 ± 0.37	98.31 ± 0.75
*n*-nonadecylic acid	29.85 ± 0.27	33.45 ± 0.30	40.20 ± 0.36	76.05 ± 0.65	29.83 ± 0.27	56.07 ± 0.42
arachidic acid	33.11 ± 0.23	-	31.18 ± 0.28	-	64.52 ± 0.53	36.69 ± 0.28
*n*-heneicosanoic acid	44.75 ± 0.38	38.68 ± 0.35	49.37 ± 0.48	-	30.53 ± 0.22	60.64 ± 0.48
behenic acid	34.48 ± 0.24	-	66.93 ± 0.57	-	53.21 ± 0.46	42.17 ± 0.38
The total readings(mg/g)	12.33 ± 0.10	11.33 ± 0.08	10.79 ± 0.09	10.64 ± 0.08	10.11 ± 0.08	11.56 ± 0.09

Data are expressed as mean value ± S.D. -. Not detected.

**Table 9 molecules-27-05443-t009:** The contents of fatty acids from *Potentilla anserina* L. by MAE (µg/g, *n* = 3).

Samples	Sample 1	Sample 2	Sample 3	Sample 4	Sample 5	Sample 6
octanoic acid	1928.02 ± 14.82	2134.463 ± 16.58	1953.98 ± 14.30	2204.09 ± 18.76	2676.68± 21.37	2561.895± 20.12
capric acid	76.13 ± 0.55	65.27 ± 0.59	-	39.74 ± 0.34	90.01 ± 0.79	85.51 ± 0.69
undecanoic acid	890.37 ± 7.15	1312.62± 10.32	1356.34± 10.73	1228.44 ± 10.32	1274.28 ± 10.27	1226.00 ± 10.81
lauric acid	-	73.30 ± 0.55	81.92 ± 0.69	37.96 ± 0.30	87.17 ± 0.63	91.82 ± 0.74
myristic acid	97.67 ± 0.73	-	75.37 ± 0.64	49.90 ± 0.46	58.31 ± 0.46	53.53 ± 0.41
α-Linolenic acid	635.28 ± 5.77	586.39 ± 0.43	604.86 ± 4.65	496.29 ± 3.65	462.04 ± 3.27	716.21 ± 5.58
linolic acid	4208.402 ± 33.74	4510.11 ± 35.19	4032.46 ± 31.93	3866.49 ± 28.72	4507.88 ± 35.57	5160.643± 36.81
pentadecanoic acid	-	59.36 ± 0.55	65.29 ± 0.57	105.05 ± 0.87	67.34 ± 0.56	59.19 ± 0.43
palmitic acid	1770.14 ± 13.15	1344.61± 10.04	1195.255 ± 9.73	1217.593 ± 9.93	1268.70 ± 10.38	1285.70 ± 10.44
oleic acid	2902.65 ± 21.32	2179.61 ± 16.35	2158.45 ± 16.24	1437.14 ± 10.99	1959.51 ± 14.23	3399.37 ± 24.41
heptadecanoic acid	51.82 ± 0.51	52.18 ± 0.47	87.77 ± 0.72	40.10 ± 0.38	43.14 ± 0.32	66.24 ± 0.54
stearic acid	63.30 ± 0.44	65.06 ± 0.54	62.25 ± 0.56	75.13 ± 0.60	75.39 ± 0.61	154.67 ± 1.26
*n*-nonadecylic acid	-	-	38.69 ± 0.37	43.63 ± 0.33	-	73.45 ± 0.60
arachidic acid	-	76.65 ± 0.66	-	-	58.14 ± 0.41	69.47 ± 0.55
*n*-heneicosanoic acid	47.69 ± 0.33	58.67 ± 0.53	69.34 ± 0.60	73.21 ± 0.61	66.71 ± 0.57	27.69 ± 0.23
behenic acid	-	22.57 ± 0.20	85.17 ± 0.75	-	-	61.77 ± 0.43
The total readings(mg/g)	12.67 ± 0.10	12.54 ± 0.09	11.87 ± 0.09	10.91 ± 0.09	12.70 ± 0.10	15.09 ± 0.11
Samples	Sample 7	Sample 8	Sample 9	Sample 10	Sample 11	Sample 12
octanoic acid	2028.07 ± 14.09	1491.75 ± 11.09	1918.03 ± 14.24	1913.80 ± 14.03	1902.47 ± 14.22	1717.26 ± 13.13
capric acid	-	26.77 ± 0.20	59.55 ± 0.45	28.74 ± 0.23	-	45.13 ± 0.37
undecanoic acid	1215.42 ± 10.12	1217.32 ± 10.53	956.79 ± 7.37	868.47 ± 7.86	1145.03 ± 9.92	967.07 ± 7.92
lauric acid	52.28 ± 0.48	-	41.15 ± 0.34	46.09 ± 0.36	51.93 ± 0.49	37.33 ± 0.29
myristic acid	-	23.93 ± 0.24	65.33 ± 0.56	30.04 ± 0.23	50.54 ± 0.44	76.23 ± 0.67
α-Linolenic acid	531.48 ± 4.78	538.09 ± 4.02	619.29 ± 5.45	385.64 ± 2.83	523.22 ± 4.42	465.51 ± 3.63
linolic acid	3507.09 ± 25.41	2101.88± 14.78	3532.46 ± 25.39	3645.67± 25.77	4554.67± 33.96	4367.58± 34.02
pentadecanoic acid	67.62 ± 0.58	29.30 ± 0.27	36.22 ± 0.32	36.07 ± 0.30	58.78 ± 0.46	20.13 ± 0.21
palmitic acid	1265.36 ± 10.38	1160.27 ± 9.91	875.65 ± 6.72	978.49 ± 7.66	845.31 ± 7.04	1017.32 ± 9.60
oleic acid	1681.14± 13.04	2007.83 ± 13.72	1979.86 ± 12.87	1269.45 ± 10.46	1578.56 ± 12.78	2015.12± 18.59
heptadecanoic acid	48.37 ± 0.46	36.38 ± 0.30	59.76 ± 0.49	49.41 ± 0.44	-	37.34 ± 0.30
stearic acid	92.60 ± 0.83	-	29.72 ± 0.27	33.76 ± 0.31	23.31 ± 0.23	50.67 ± 0.39
*n*-nonadecylic acid	57.09 ± 0.50	25.60 ± 0.25	45.99 ± 0.35	45.90 ± 0.36	24.67 ± 0.19	41.45 ± 0.29
arachidic acid	-	-	31.81 ± 0.27	-	37.14 ± 0.30	29.63 ± 0.24
*n*-heneicosanoic acid	57.61 ± 0.52	42.02 ± 0.34	35.80 ± 0.27	-	30.47 ± 0.24	33.93 ± 0.31
behenic acid	41.16 ± 0.32	-	52.43 ± 0.46	-	36.89 ± 0.30	51.90 ± 0.44
The total readings(mg/g)	10.65 ± 0.08	8.70 ± 0.07	10.34 ± 0.08	9.33 ± 0.07	10.86 ± 0.08	10.97 ± 0.09

Data are expressed as mean value ± S.D. -. Not detected.

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
