# Peer review of "Optimization and Comparative Study of Different Extraction Methods of Sixteen Fatty Acids of *Potentilla anserina* L. from Twelve Different Producing Areas of the Qinghai-Tibetan Plateau"

_molecules, 2022, doi:10.3390/molecules27175443_

Round 1

Reviewer 1 Report

As a general comment, this work has good information content. However, the terminology for supercritical fluid extraction is SFE or supercritical carbon dioxide scCO2. These terminologies are universally accepted, in this way keep the common terminology instead use another one. In this way, please change the acronyms SCDE and use SFE indicating that the authors used carbon dioxide as the solvent.

In Table 1, please separate the picking time data to better see the year, month and day. About the picking time, what is the age of those plants? Is very important to guarantee that the plants were at the same growing time since it will conditioner the production of the metabolites.

For information about the extraction variables, some of them must be similar to compare results. If all variables are different and the mechanisms of extraction are different, there is no comparison possible among extraction methods.

Please, identify what extraction method was developed in continuous or static mode.

For SFE is missing information about the flow rate of solvent and the solid/solvent ratio.

For UAE the solid/solvent ratio is identified as 7.3:1 ml/g, but it is not the same for UAE and SFE. Authors cannot compare the extractions in this way.

About the used solvent for UAE, why manganese chloride solution, glycerin, pyridoxal phosphate and magnesium chloride solution were added to the solvent? There is no explanation for this, and it is difficult the analysis this method when the solvent is radically changed.

For section 2.5 to prepare the standards why 2-(4-amino)-phenyl-1-hy-132 drogen-phenanthrene [9,10-d] imidazole, 4 - dimethyl aminopyridine and 1- ethyl - 3 - (3 - dimethyl ammonium 139 propyl) - carbonyl imine were added in the solution? Similar to the last comment, there is no explanation for this. What kind of reaction authors want to promote?

For the HPLC method, please present the acetonitrile and water concentration separated and not in solution.

For the 3.1 section authors must improve the discussion about the extraction processes there is no reason to compare the 3 methods. Please, merge this section with section 3.4 and improve this discussion by highlighting the differences among methods and variables that could influence the extraction of fatty acids.

In the 3.2 section, there are cited many columns used here, but in the materials and method section, there is only one. If this information about the comparison of the column doesn’t matter, remove this description (lines 212-215).

About the Figure 2. What is a blank chromatogram? There is a peak PIA, what is this?

Please, present a chromatogram for each extraction process, and format the botanical plant's name in italic. In the same figure 2 title, the chromatogram is for Potentilla anserina L extracts, and not of the plant, since this chemical profile depends on the solvent used.

The title of the article proposed to study the fatty acids extraction from different producing areas, however, the authors did not relate the composition with these different producing areas. A generic discussion is made in the last section, lines 283-294, and this discussion must be improved to relate the edaphoclimatic characteristics of each producing area.

Author Response

Reviewer 1: Comments and Suggestions for Authors

As a general comment, this work has good information content. However, the terminology for supercritical fluid extraction is SFE or supercritical carbon dioxide scCO2. These terminologies are universally accepted, in this way keep the common terminology instead use another one. In this way, please change the acronyms SCDE and use SFE indicating that the authors used carbon dioxide as the solvent.

Thanks for your professional suggestion. According to your advice, we changed the acronyms SCDE and used SFE. We think this may be more accurately than the previous one. (page 1, line 12; line 29; page 2, line 50; page 4, line 109; page 15, line 316 and so on)

In Table 1, please separate the picking time data to better see the year, month and day. About the picking time, what is the age of those plants? Is very important to guarantee that the plants were at the same growing time since it will conditioner the production of the metabolites.

Thanks for your careful suggestion. According to your advice, we separated the picking time data to better see the year, month and day. In addition,all samples were propagated by cuttings in 2018 and were 3 years old at the time of collection. In response to your suggestion, we have added this part of the description to the manuscript (page 3, Table 1; page 2, line 71-72)

For information about the extraction variables, some of them must be similar to compare results. If all variables are different and the mechanisms of extraction are different, there is no comparison possible among extraction methods.

We would like to express our sincere thanks for your valuable suggestions on our manuscript. In fact, our response surface optimization extraction technology was based on single factors experiment. The key total fatty acids extraction parameters by SFE were optimized:Xa1, extraction temperature; Xa2, extraction pressure; Xa3, extraction time and Xa4, flow of carbon dioxide. The important parameters affecting total fatty acids extraction from UAE were optimized:Xb1, extraction temperature; Xb2, extraction time; Xb3, extraction volume and Xb4, extraction times. The critical characteristics affecting total fatty acids extraction by MAE were optimized:Xc1, extraction power; Xc2, extraction time; Xc3, extraction times and Xc4, extraction volume. The extraction time was all the significant factor affecting the extraction yield in three different extraction methods. The other two factors which had the greatest influence on the yield of fatty acid were further optimized by response surface method. Therefore, the view influencing factors of each extraction method may be different, which is inevitable in the optimization experiment. But the purpose of this study is to get the most suitable extraction method for fatty acids of Potentilla anseris L. by the experiment comparison of fatty acid extraction rate not the influencing factor of the experimental process.

Please, identify what extraction method was developed in continuous or static mode.

Thanks for your careful suggestion. In this study, SFE, UAE and MAE were all developed in static mode. In response to your valuable suggestion, we have added this part of the description to the manuscript (page 3, line 97-98)

For SFE is missing information about the flow rate of solvent and the solid/solvent ratio.

Thanks for your professional suggestion. According to your advice, we added the information about the flow rate of solvent: flow of carbon dioxide was 40 L/h. In addition, because carbon dioxide was used as the extraction solvent in the extraction process, the solid-liquid ratio is not involved (page 4, line 113)

For UAE the solid/solvent ratio is identified as 7.3:1 ml/g, but it is not the same for UAE and SFE. Authors cannot compare the extractions in this way.

We would like to express our sincere thanks for your valuable suggestions on our manuscript. The solid/solvent ratio of UAE and MAE were not the same and because carbon dioxide was used as the extraction solvent in SFE, the solid-liquid ratio is not involved. Moreover, the most critical factor in the experiment was comparisons of fatty acid extraction rate to get the most suitable extraction method for fatty acids of Potentilla anseris L. rather not the comparison of the solid/solvent ratio that influencing factor of the experimental process.

About the used solvent for UAE, why manganese chloride solution, glycerin, pyridoxal phosphate and magnesium chloride solution were added to the solvent? There is no explanation for this, and it is difficult the analysis this method when the solvent is radically changed.

Thanks for your valuable suggestions. We're sorry for the slip of pen here and the extraction reagent used in the experiment was methanol-methylene chloride solution (1:3, V/V). The UAE of total fatty acids from Potentilla anserina L was carried out according to the existing literatures[1-3] report with minor changes. We have made changes in the manuscript and checked carefully to make sure there are no such low-level errors. (page 4, line 118-119)

  1. Quancai P., Jinming S., Jun L., Ying Y.; Determination of 30 Fatty Acids in Scallop by Ultrasonic Extraction Gas Chromatography. Food science, 2012, Vol. 33, No. 12
  2. Farid, C.; Natacha, R.; Anne-Gaëlle, S.; Alice, M.; Anne-Sylvie, F.; Maryline, A., Ultrasound assisted extraction of food and natural products. Mechanisms, techniques, combinations, protocols and applications. A review. Ultrasonics - Sonochemistry 2017, 34.
  3. Brijesh, K.T., Ultrasound: A clean, green extraction technology. Trends in Analytical Chemistry 2015, 71.

For section 2.5 to prepare the standards why 2-(4-amino)-phenyl-1-hy-132 drogen-phenanthrene [9,10-d] imidazole, 4 - dimethyl aminopyridine and 1- ethyl - 3 - (3 - dimethyl ammonium 139 propyl) - carbonyl imine were added in the solution? Similar to the last comment, there is no explanation for this. What kind of reaction authors want to promote?

Thanks for your professional suggestion. That's a really good question. Determination of fatty acids can be challenging because they are not volatile, do not absorb UV or visible light and do not have fluorescence properties. Thus, the most common methods used for their analysis are liquid chromatography (LC) or gas chromatography (GC) [1,2] coupled with derivatization or gas chromatography–mass spectrometry (GC–MS) [3,4]. Compared with GC methods, the HPLC analysis of fatty acids can be carried out under mild conditions. Thus, the risks of damaging heat-liable compounds are greatly reduced. In addition, the detectors used in HPLC are nondestructive for the analytes, which makes it possible to collect the fraction for further investigation; Therefore, HPLC method has developed rapidly to determine the content of fatty acids because of the fast analysis speed, good selectivity and high separation efficiency. At present, post-column derivatization is a well-accepted method to enhance the selectivity and sensitivity for analysis of fatty acids. In this study, as the pre-column derivatization reagent, 2-(4-amino)-phenyl-1-hydrogen-phenanthrene [9, 10-d] imidazole has been used for the sensitive and selective determination of saturated and unsaturated fatty acids. 4 - dimethyl aminopyridine (DMAP) and 1- ethyl - 3 - (3 - dimethyl ammonium propyl) - carbonyl imine (EDC) were all necessary reagents in derivatization scheme for the representative fatty acid as shown in next Fig. In addition, detailed information about the derivatization reagents and the derivatization process is reported in our previous article [5].

  1. Takahashi K, Goto-Yamamoto N (2011) Simple method for the simultaneous quantification of medium-chain fatty acids and ethyl hexanoate in alcoholic beverages by gas chromatography-flame ionization detector: Development of a direct injection method. J Chromatogr A 1218:7850-7856
  2. Soliman LC, Andrucson EM, Donkor KK, Church JS, Cinel B (2015) Determination of Fatty Acids in Beef by Liquid Chromatography-Electrospray Ionization Tandem Mass Spectrometry. Food Anal. Methods 9: 630-637
  3. Quehenberger O, Armando AM, Dennis EA (2011) High sensitivity quantitative lipidomics analysis of fatty acids in biological samples by gas chromatography-mass spectrometry. Biochim Biophys Acta 1811: 648-656
  4. Manzano P, Diego JC, Nozal MJ, Bernal JL, Bernal J (2012) Gas chromatography–mass spectrometry approach to study fatty acid profiles in fried potato crisps. J Food Compos Anal 28: 31-39
  5. Wang, Y, Luan, G, Zhou, W, You, J, Hu, N, & Suo, Y. (2018). 2-(4-amino)-phenyl-1-hydrogen-phenanthrene [9,10-d] imidazole as a novel fluorescent labeling reagent for determination of fatty acids in raspberry. Food Analytical Methods.

For the 3.1 section authors must improve the discussion about the extraction processes there is no reason to compare the 3 methods. Please, merge this section with section 3.4 and improve this discussion by highlighting the differences among methods and variables that could influence the extraction of fatty acids.

Thanks for your helpful suggestion. The purpose of 3.1section was optimal the best SFE, UAE and MAE conditions by optimize a series of designed extraction variables and explore the interactions between these variables by the BBD. The purpose of 3.4 section was comparison of SFE, UAE and MAE to obtain the best extraction method. These two parts were independent of each other in the previous similar literatures[1,2]. We have followed your advice to move Table 2-4 (The experimental design and results of fatty acids) in Results and discussion. After revision, the manuscript was more structured and transparent. In addition, according to your opinion,we improved section 3.4 discussion by highlighting the differences among methods with the results of other authors' (page 5, Table 2-4 and page 11, line 267-280)

  1. Halil, O., Ilkay, K., (2016) Application of response surface methodology for optimizing the recovery of phenolic compounds from hazelnut skin using different extraction methods. Industrial Crops and Products 91:114–124
  2. Margeretha, I., Suniarti, D. F., Herda, E., & ZA Mas’Ud. (2012). Optimization and comparative study of different extraction methods of biologically active components of indonesian propolis trigona spp. Journal of Natural Products, Vol. 5: 233- 242.

In the 3.2 section, there are cited many columns used here, but in the materials and method section, there is only one. If this information about the comparison of the column doesn’t matter, remove this description (lines 212-215).

Thanks for your valuable suggestions. The information about the comparison of the column does really important in optimization of HPLC separation, we have added columns information in the materials and method section. Thank you very much for your critical comments on this section, which we have indeed neglected in the manuscript. (page 3, line 86-90 and page 9, line 215-217)

About the Figure 2. What is a blank chromatogram? There is a peak PIA, what is this?

Thanks for your careful suggestion. The blank chromatogram is a chromatogram obtained using only the derived reagent. In this study, the pre-column derivatization reagent was 2-(4-amino)-phenyl-1-hydrogen-phenanthrene [9, 10-d] imidazole. It was simply called PIA in Figure 2. We have added remarks in the manuscript (page 9, line 230-231)

Please, present a chromatogram for each extraction process, and format the botanical plant's name in italic. In the same figure 2 title, the chromatogram is for Potentilla anserina L extracts, and not of the plant, since this chemical profile depends on the solvent used.

We would like to express our sincere thanks for your valuable suggestions on our manuscript. We have followed your advice to present chromatograms for each extraction process, format the botanical plant's name in italic and modify figure 2 title. (page 9, line 223-227 and figure 2)

The title of the article proposed to study the fatty acids extraction from different producing areas, however, the authors did not relate the composition with these different producing areas. A generic discussion is made in the last section, lines 283-294, and this discussion must be improved to relate the edaphoclimatic characteristics of each producing area.

Thanks for your professional suggestion. We have followed your advice to improve discussion to relate the fatty acids composition with different producing areas. As shown in Table 7-9,the total fatty acids contents of Potentilla anserina L. growing in Gannan Tibetan Autonomous Prefecture was as higher to 18.11 ± 0.13 mg/g and the total fatty acids contents of sample growing in Anduo County Tibet Autonomous Region was only 8.70 ± 0.07 mg/g. According to Table 1,the altitude of Gannan Tibetan Autonomous Prefecture and Anduo County Tibet Autonomous Region was 2950m and 4600m, respectively and the coordinates of Gannan Tibetan Autonomous Prefecture and Anduo County Tibet Autonomous Region was 33°06′37.49′′N, 100°46′12.77′′E and 31°40′14.31′′N, 91°51′23.78′′E, respectively. The samples were planted artificially so factors that can be controlled by researchers was basically the same. The differences of fatty acids of Potentilla anserina L. from different habitats maybe related to local altitude. In addition, the study did not analyze the differences resulting from the edaphoclimatic characteristics of each producing area. After discussion among the authors, we deleted this part in the conclusions. (page 14, line 305-314).

The authors really appreciate your carefully reading and comments. All of the mistakes mentioned above have been corrected in the revised manuscript. At last, plenty of thanks to you again for the valuable comments for our paper. We learned much from your advice.

Reviewer 2 Report

“Quantitative Analyses of 16 Fatty Acids in Potentilla anserina L. from the Qinghai-Tibetan Plateau Different Producing Areas”

At the moment, the publication cannot be approved. The manuscript needs to be completely revised. The main topic of that manuscript is comparison of the extraction method of fatty acids from Potentilla anserina L., so the title should be connected with the main topic. Methodologies and results are intertwined with each other, it should be more structured and transparent. The results are not compared with other publications.

There is no information which part of the Potentilla anserine L you use for fatty acids extraction?

Line 2 - The font size should be the same

Line 11 – Delete space (in manuscript there are a lot of double space)

Line 18-19 – Why sample no 6 was chosen?

Line 19-20 The results should be calculated to mg/g

Line 29 – Why “fatty acids” is capitalized?

Line 64, 79 – Should be “linoleic acid” in all places in manuscrpit

Line 72 – Professor Yongfang Li name is presented with other authors, so here it isn’t important to show that name. It’s better if you show how Potentilla anserina L. were identified?

Line 93-94 Why the samples from Yushu Tibetan Autonomous Prefecture were choose?

Line 102 – Delete “min”

Line 103 – “total peak area of fatty acids”

Line 112 – How you homogenate Potentilla anserine L

Line 141 - “0.600 mL acetonitrile solution (?) was added then the diluted solution was syringe filtered.” Are you sure that was acetonitrile solution? Isn’t it pure acetonitrile?

Figure 2 I think it isn’t important to show blank sample

Line 222 – There should be italic name of the plant

Line 131 – There is no reference connected to the preparation of sample and HPLC analysis

Line 211 – Delete “obtain”

Table 7-9 The results should be shown in mg/g

Table 2-4 Peak area should be in Results

Line 285-294 In that manuscript there is no correlation with climatic factors, so you can’t wrote about it.

Why are there different chromatograms in the Original images?

Line 374 – Should be corrected

Author Response

Reviewer 2: Comments and Suggestions for Authors

“Quantitative Analyses of 16 Fatty Acids in Potentilla anserina L. from the Qinghai-Tibetan Plateau Different Producing Areas”

At the moment, the publication cannot be approved. The manuscript needs to be completely revised. The main topic of that manuscript is comparison of the extraction method of fatty acids from Potentilla anserina L., so the title should be connected with the main topic. Methodologies and results are intertwined with each other, it should be more structured and transparent. The results are not compared with other publications.

Thanks for your professional suggestion. According to your suggestion, we changed the title to: Optimization and Comparative Study of Different Extraction Methods of 16 Fatty Acids in Potentilla anserina L. from the Qinghai-Tibetan Plateau Different Producing Areas. The title was better connected with the main topic of that manuscript with modification. We have followed your advice to adjusted the layout of Methodologies and results that move Table 2-4 (The experimental design and results of fatty acids) in Results and discussion. After revision, the manuscript was more structured and transparent. In addition, according to your opinion,we improved discussion to highlight the differences among methods by compared with other publications. (page 1, line 1-3; page 5, Table 2-4 and page 11, line 267-280)

There is no information which part of the Potentilla anserine L you use for fatty acids extraction?

Thanks for your careful suggestion. The tuberous root of Potentilla anserine L. was used for fatty acids extraction in this study. We have added these informations of sample in the manuscript (page 2, line 70-71)

Line 2 - The font size should be the same

Thanks for your careful suggestion. We've resized the fonts to make them exactly the same in the manuscript (page 1, line 3)

Line 11 – Delete space (in manuscript there are a lot of double space)

Thanks for your valuable suggestion. We've deleted all double space in the manuscript (page 1, line35; page 2, line44; page 4, line 103 and 106; page 6, line 173; page 7, line190; page 15, line 324; line336 and so on)

Line 18-19 – Why sample no 6 was chosen?

Thanks for your professional suggestion. By this study, the results show that amounts of total fatty acids in sample no 6 by SFE, UAE and MAE were 16.58 ± 0.14 mg/g, 18.11 ± 0.13 mg/g and 15.09 ± 0.11 mg/g, respectively. The total content of fatty acids and the monomer content of 16 fatty acids in the sample no 6 were significantly higher than those in the samples from other producing areas. Then sample no 6 was chosen.

Line 19-20 The results should be calculated to mg/g

Thanks for your valuable suggestion. The results have been calculated to mg/g in the manuscript (page 1, line 20; page 11, line 254-257; page 14, line 297, 298, 306, 308; page 15, line 325-328and table 7-9)

Line 29 – Why “fatty acids” is capitalized?

Thanks for your careful suggestion. We have removed capitalized of fatty acid and changed it to lowercase in the manuscript (page 1, line 29)

Line 64, 79 – Should be “linoleic acid” in all places in manuscrpit

Thanks for your professional suggestion. The introduction of 16 fatty acids appeared in many parts of the manuscript which was really cumbersome. According to your suggestion,we have deleted it (page 2, line 64-66)

Line 72 – Professor Yongfang Li name is presented with other authors, so here it isn’t important to show that name. It’s better if you show how Potentilla anserina L. were identified?

Thanks for your valuable suggestion. According to your suggestion,we have deleted Professor Yongfang Li name. We identify Potentilla anserina L. mainly on the basis of plant appearance. In addition, all samples were propagated by cuttings in 2018 by our team. (page 2, line 70-71)

Line 93-94 Why the samples from Yushu Tibetan Autonomous Prefecture were choose?

Thanks for your careful suggestion. There is a slip of the pen here and we have modified it in the manuscript. The samples from Gannan Tibetan Autonomous Prefecture was selected as the representative one (page 3, line 96-97)

Line 102 – Delete “min”

Thanks for your careful suggestion. We have deleted “min” in the manuscript (page 4, line 106)

Line 103 – “total peak area of fatty acids”

Thanks for your valuable suggestion. According to your suggestion,we have modified to “total peak area of fatty acids” in the manuscript.  (page 4, line 106)

Line 112 – How you homogenate Potentilla anserine L

Thanks for your careful suggestion. It should be described by powder not homogenate and we have modified it in the manuscript. (page 4, line 110,117 and 121)

Line 141 - “0.600 mL acetonitrile solution (?) was added then the diluted solution was syringe filtered.” Are you sure that was acetonitrile solution? Isn’t it pure acetonitrile?

We really appreciate your carefully reading. We checked the records of previous experiments to make sure that was pure acetonitrile solution which completely consistent with our previous literature[1].

  1. Wang, Y, Luan, G, Zhou, W, You, J, Hu, N, & Suo, Y. (2018). 2-(4-amino)-phenyl-1-hydrogen-phenanthrene [9,10-d] imidazole as a novel fluorescent labeling reagent for determination of fatty acids in raspberry. Food Analytical Methods.

Figure 2 I think it isn’t important to show blank sample

Thanks for your professional suggestion. We have deleted blank sample in Figure 2 and presented chromatograms for each extraction process (Figure 2)

Line 222 – There should be italic name of the plant

Thanks for your valuable suggestion. According to your suggestion, we format the botanical plant's name in italic and modify figure 2 title. (page 9, line 223-227 and figure 2)

Line 131 – There is no reference connected to the preparation of sample and HPLC analysis

Thanks for your careful suggestion. According to your suggestion, we have added reference connected to the preparation of sample and HPLC analysis in the manuscript. As a new derivative reagent ,2-(4-amino)-phenyl-1-hydrogen-phenanthrene [9,10-d] imidazole has been used to HPLC analysis fatty acids only once in the available literature. (page 4, line 134 and 140)

Line 211 – Delete “obtain”

Thanks for your valuable suggestion. According to your suggestion, we deleted “obtain”. (page 9, line 214)

Table 7-9 The results should be shown in mg/g

Thanks for your professional suggestion. The total readings have been changed to shown by mg/g. The content of some fatty acids a little low, such as the content of arachidic acid in sample 7 by SFE was only 21.38±0.37 mg/g. If the results been shown in mg/g and accurated to two decimal places, it was 0.02±0.00 ug/g. The total readings should be shown by mg/g and the content of some fatty acids was better to shown in ug/g. (Table 7-9)

Table 2-4 Peak area should be in Results

Thanks for your valuable suggestions. We have followed your advice to move Table 2-4 in Results and discussion. (page 5, Table 2-4).

Line 285-294 In that manuscript there is no correlation with climatic factors, so you can’t wrote about it.

Thanks for your professional suggestion. According to your suggestion, we deleted this part in the conclusions and improve discussion to relate the fatty acids composition with different producing areas. (page 14, line 305-314).

Why are there different chromatograms in the Original images?

Thanks for your valuable suggestion. The wrong chromatograms were applied during the insertion of the manuscript. We have corrected the mistake. (Figure 2)

Line 374 – Should be corrected

Thanks for your careful suggestion. According to your suggestion, we have corrected reference. (page 16, line 397)

We are very grateful to your carefully reading and comments for the manuscript. All of the mistakes mentioned above have been corrected in the revised manuscript. At last, plenty of thanks to you again for the valuable comments for our paper. We learned much from your advice.

Reviewer 3 Report

The manuscript presented for revision is very interesting. This work is well organized and scientifically sound. However, the authors did not avoid some minor mistakes that we all make while preparing the publications.

In the introduction, it is possible to further describe the fat fraction of Potentilla anserina L., to characterize the composition of fatty acids and other bioactive compounds. The value of this paper could also be increased by a comparison of the obtained research results with the results of other authors' discussion of the results (e.g. Xia, L., Song, C., Sun, Z. et al. Determination of Free Fatty Acids in Tibet Folk Medicine Potentilla anserina L. Using a New Labeling Reagent by LC with Fluorescence Detection and Identification with Online Atmospheric Chemical Ionization-MS Identification. Chroma 71, 623-631 (2010). Https://doi.org/10.1365/s10337-010-1523-z) - although I know it won't be easy.

n the conclusions, the authors refer to the differences resulting from cultivation conditions, which were not analyzed in this work, their influence on the fatty acid profile was not assessed. I propose to delete this sentence (line 319-321).

Author Response

Reviewer 3: Comments and Suggestions for Authors

The manuscript presented for revision is very interesting. This work is well organized and scientifically sound. However, the authors did not avoid some minor mistakes that we all make while preparing the publications.

Thank you very much for your appreciation to our work. Thank you again for your review! According with your advice, we amended the relevant part in manuscript. Some of your questions were answered below. All revisions have been clearly marked in the electronic version of the revised manuscript by using a red background.

In the introduction, it is possible to further describe the fat fraction of Potentilla anserina L., to characterize the composition of fatty acids and other bioactive compounds. The value of this paper could also be increased by a comparison of the obtained research results with the results of other authors' discussion of the results (e.g. Xia, L., Song, C., Sun, Z. et al. Determination of Free Fatty Acids in Tibet Folk Medicine Potentilla anserina L. Using a New Labeling Reagent by LC with Fluorescence Detection and Identification with Online Atmospheric Chemical Ionization-MS Identification. Chroma 71, 623-631 (2010). Https://doi.org/10.1365/s10337-010-1523-z) - although I know it won't be easy.

Thanks for your constructive suggestion. We have added further describe the fat fraction of Potentilla anserina L. in the introduction and comparison of the obtained research results with the results of other authors' (page 2, line 55-58 and page 11, line 267-280)

in the conclusions, the authors refer to the differences resulting from cultivation conditions, which were not analyzed in this work, their influence on the fatty acid profile was not assessed. I propose to delete this sentence (line 319-321).

Thanks for your professional suggestion. According to your advice, we deleted this sentence. (page 14, line 305-314)

At last, plenty of thanks to you again for the valuable comments for our paper If you have any question about this paper, please don’t hesitate to contact us. We learned much from your advice.

Round 2

Reviewer 1 Report

Dear authors 

In general, this manuscript was improved, including the methods descriptions, and the doubts were clarified.

Authors carefully answered each comment, and now this version is suitable to be published.

Author Response

Review Comments

Reviewer 1: Dear authors

In general, this manuscript was improved, including the methods descriptions, and the doubts were clarified.

Authors carefully answered each comment, and now this version is suitable to be published.

Thank you very much for your appreciation to our work! We have learned a lot from your constructive comments, helpful suggestions and detailed instructions. Thank you again sincerely.

Sincerely yours,

Yongfang Li

Reviewer 2 Report

Line 19 – should be „sample 6”

Line 111, 117, 121 – should be „10 g”

Figure 1 – the font must be bigger

Figure 2 – numbers are also too small

Line 256 – here should be a reference

Line 267-280 it should be deleted. You didn’t show Your detailed amounts of each fatty acids.

Line 307 – should be a reference

Line 308 – which method was used for extraction?

In discussion should be more references, for examples some comparison with other publications connected with different method of lipids extractions.

Author Response

Reviewer 2: Comments and Suggestions for Authors

Thank you for your review! According with your advice, we amended the relevant part in manuscript. Some of your questions were answered below. All revisions have been clearly marked in the electronic version of the revised manuscript by using a yellow background.

Line 19 – should be „sample 6”

Thanks for your careful suggestion. According to your advice, we have changed to “sample 6”. (page 1, line 19)

Line 111, 117, 121 – should be „10 g”

Thanks for your professional suggestion. According to your advice, we have changed to “10 g”. (page 4, line 111, 117, 121)

Figure 1 – the font must be bigger

Thanks for your valuable suggestions. According to your advice, we have changed to bigger font. (Figure 1)

Figure 2 – numbers are also too small

Thanks for your valuable suggestions. According to your advice, we have changed to bigger numbers. (Figure 2)

Line 256 – here should be a reference

We would like to express our sincere thanks for your valuable suggestions on our manuscript. The informations of line 256 (amounts of total fatty acids in sample from Yushu Tibetan Autonomous Prefecture Qinghai by SFE, UAE and MAE) was the result of this research not from reference. In order to avoid unnecessary ambiguity, we have made appropriate corrections in the article. (page 10, line 255)

Line 267-280 it should be deleted. You didn’t show Your detailed amounts of each fatty acids.

Thanks for your constructive suggestion. According to your advice, we have deleted unnecessary information about detailed amounts of each fatty acids. We have added further comparison with other publications connected with the total contents of the free fatty acids in Potentilla anserina L. (from Yushu Tibetan Autonomous Prefecture Qinghai) by different extractions method (distillation extraction with n-hexane). (page 11, line 267-275)

Line 307 – should be a reference

We would like to express our sincere thanks for your valuable suggestions on our manuscript. The informations of line 302 (the total fatty acids contents of Potentilla anserina L. growing in Gannan Tibetan Autonomous Prefecture and Lhatse County Tibet Autonomous Region) was also the result of this research not from reference. In order to avoid unnecessary ambiguity, we have annotated the manuscript with a yellow background. (page 14, line 300-303)

Line 308 – which method was used for extraction?

Thanks for your valuable suggestions. According to your advice, we have added method that used for extraction: UAE. Before we change it, we compared the total fatty acids contents of Potentilla anserina L. growing in Gannan Tibetan Autonomous Prefecture by UAE and the total fatty acids contents of sample growing in Anduo County Tibet Autonomous Region by MAE, which one is the highest total fatty acids contents and another is the minimum one in this work. After modification, we compared the highest total fatty acids contents and minimum one by the same extraction method (UAE). The authors agreed that the comparisons were more precise and accurate after the revision in accordance with your comments. (page 14, line 291-294)

In discussion should be more references, for examples some comparison with other publications connected with different method of lipids extractions.

Thanks for your constructive suggestion. We have added further comparison with other publications connected with different extraction method for free fatty acids in Potentilla anserina L. Unfortunately, there was very limit of article that focus on fatty acid composition of Potentilla anserina L. by different extraction method and samples which also from same areas. (page 11, line 267-275)

At last, plenty of thanks to you again for the valuable comments for our paper. If you have any question about this paper, please don’t hesitate to contact us. We learned much from your advice.

Sincerely yours,

Yongfang Li